# Towards a comprehensive view of the pocketome universe—biological implications and algorithmic challenges

Hanne Zillmer, Dirk Walther ⬡ *

Max Planck Institute of Molecular Plant Physiology, Potsdam-Golm, Germany

* walther@mpimp-golm.mpg.de

## Abstract

With the availability of reliably predicted 3D-structures for essentially all known proteins, characterizing the entirety of compound-binding sites (binding pockets on proteins) has become a possibility. The aim of this study was to identify and analyze all compound-binding sites, i.e., the pocketomes, of eleven species from different kingdoms of life to discern evolutionary trends as well as to arrive at a global cross-species view of the pocketome universe. Computational binding site prediction was performed on all protein structures in each species as available from the Alpha-Fold database. The resulting set of potential binding sites was inspected for overlaps with known pockets and annotated with regard to the protein domains in which they are located. 2D-projection plots of all pockets embedded in a 128-dimensional feature space, and characterizing them with regard to selected physicochemical properties, provide informative, global pocketome maps that unveil differentiating features between pockets. Our study revealed a sub-linear scaling law of the number of unique binding sites relative to the number of unique protein structures per species. Thus, as proteomes increased in size during evolution and therefore potentially diversified, the number of distinct binding sites, reflecting potentially diversifying functions, grew less than proportionally. We discuss the biological significance of this finding as well as identify critical and unmet algorithmic challenges.

## Author summary

The function of proteins is governed by specific interactions with other molecules, notably small molecules (compounds, such as metabolites). The precise nature of the protein-compound interaction, and thus, the associated function, is determined by the stereochemical and physicochemical properties of the sites at which the interaction occurs (binding pockets). Thus, novel functions (binding of novel compounds) generally require the emergence of new binding sites. With the recent breakthroughs in protein structure prediction, the complete set of

**Data availability statement:** The software code developed and/ or used to conduct this study, along with corresponding input files, is available at https://github.com/zillmerhanne/PocketomeUniverse. In addition, this repository contains all pocket descriptors and embeddings produced by DeeplyTough, as well as interactive Plotly plots for the tSNE projections of the DeeplyTough embeddings, which allow filtering for specific compound classes or species.

**Funding:** The author(s) received no specific funding for this work.

**Competing interests:** The authors have declared that no competing interests exist.

protein structures has become available. This allowed us to apply computational binding site predictions and to investigate the entirety of all pockets (the "pocketome") across eleven species from different kingdoms of life, and to study the relationship between the emergence of novel binding sites in relation to increasing sizes of proteomes, i.e., the set of all protein structures in a given species. Our analysis uncovered a sub-linear relationship between the numbers of unique pockets and unique protein structures, suggesting that during evolution, functional diversity shows signs of saturation, which is consistent with other reports, but approached here from the perspective of compound-binding specificities. Our study constitutes the first large-scale investigation of pocketomes based on the now available high-confidence protein structures.

## Introduction

To perform their biochemical function, proteins interact with various entities such as small molecules, generally referred to as compounds or specifically in biological systems as metabolites. These interactions occur at cavities on the protein surface known as "binding pockets" or "compound binding sites". Binding pockets have been the subject of both experimental and bioinformatic research for many years, with the aim to assess similarities between them, to identify potential ligands for specific binding sites, or all potential target proteins for a given compound. A binding pocket is typically defined as all residues within a certain distance to the ligand's heavy atoms in a given protein-ligand complex [1]. As this definition requires detailed 3D-structural information of protein-compound complexes, i.e., the position and conformation of the ligand needs to be known, computational approaches have been developed to predict binding pockets from protein 3D-structures alone, i.e., in the absence of a bound ligand. Some of these approaches are geometry-based [2,3], while others are energy-based [4,5] or machine-learning-based [6,7]. With the recent advances in protein structure prediction and the release of the AlphaFold database, these approaches allow to predict and analyze potential binding sites for whole proteomes of the covered 48 different and evolutionarily diverse species, such as *Arabidopsis thaliana* or human. Thus, a comprehensive and systematic view of compound binding sites on protein surfaces has now become possible.

In this study, we aimed to systematically characterize, cluster, and compare complete sets of protein-compound binding sites. As in the case of small compound ligands, binding sites are frequently referred to as binding pockets, we essentially took aim at describing whole "pocketomes" of different species. By taking a cross-species view and including species of different evolutionary history (bacteria, fungi, plants, mammals), we wished to understand how the pocketomes evolved along with different metabolic needs and organismic capabilities. In particular, as species differ markedly with regard to their proteomes, notably by the number of encoded proteins, we wished to discern the corresponding scaling law that relates the number of unique pockets, reflecting on the potential to bind a corresponding

number of unique compounds, to the number of proteins encoded in the respective species. Such an analysis may shed light on the trends of evolution with regard to "investment", meaning whether binding capacities scale proportionally with proteome size, or emerged at increased or decreased rates, with all possible outcomes being relevant for an understanding of selection processes in evolution.

Each species has a unique set of proteins and, thus, binding pockets that is involved in metabolic interactions as well as signaling processes. These binding pockets collectively define the species' pocketome. Thus far, only a few studies on pocketomes have been performed. These include studies on the pocketome of certain enzyme classes, such as human kinases [8] and G-protein-coupled receptors [9], the pocketome of known 3D structures [10–12], and the species-specific pocketomes, such as *Mycobacterium tuberculosis* [13] and human [14,15]. Most of these studies focused on applications in the field of biomedical research and the druggability of binding sites or discussed differences between drug-protein vs. metabolite-protein binding events [16,17]. An inter-species perspective that exploits the new scale at which structural data are available and that aims at deducing evolutionary trends has not been presented yet.

To computationally compare and cluster binding sites, they have to be represented in a machine-readable format, i.e., in a numeric format. Binding site descriptions have been developed at different levels of detail, ranging from coarse-grained representations, where each amino acid residue is represented by a single atom, e.g., the $C_a$ atom, to full-atom representations, where all atoms of the binding site are used [18]. Other approaches represent binding pockets not based on residue positions but rather as a point cloud [3]. In addition to 3D coordinates, binding pocket representations often include additional information, such as the type of residues, thereby aiming to capture the physicochemical properties. Other methods, such as DeeplyTough [19], employ deep learning techniques to learn a representation for a protein binding pocket as a numeric vector of a chosen dimensionality (referred to as "embedding").

Applying established computational methods that allow for an analysis at different levels of structural detail, our study revealed a sub-linear scaling law of the number of unique binding sites relative to the number of unique protein structures per species. Thus, larger proteomes harbor less than proportionally more different binding sites than species with smaller proteomes. We also identified and discuss challenges encountered during the process of performing a global, multi-species comparative computational pocketome analysis, such as the speed and specificity of binding site comparison algorithms, and accounting for co-factors, structural flexibility, and protein-protein interactions.

## Materials and methods

### Dataset

Predicted 3D-protein structures of 11 different species covering all kingdoms of life (excluding *Archaea*) were downloaded from the AlphaFold (AF) database v4 (https://alphafold.ebi.ac.uk/) (Table 1) [20]. All predicted structures of all species were considered for compound binding site predictions. Approximately 34% of the investigated protein structures contain at least one transmembrane segment. To ensure a high quality of proteins with alpha-helical transmembrane regions, evaluation files from the TmAlphaFold Transmembrane Protein Structure Database (https://tmalphafold.ttk.hu/) were downloaded [21]. All proteins passing less than six quality checks (as reported by the database) were excluded from further analysis, as they are classified as poor (if five quality checks are passed) and failed (if less than five quality checks are passed) models. In addition, all proteins in the AF database that are marked as fragments in UniProt and all proteins with fewer than 100 amino acid residues were excluded from further analysis. Regardingthe latter, P2Rank has been reported to underpredict on proteins shorter than 100 amino acid residues [22].

Additional information, such as known binding sites and domains, was obtained from UniProt [23] and InterPro [24], respectively. The assigned FoldSeek (FS) cluster for each protein was obtained from https://cluster.foldseek.com/ [25].

All ligands found in known (annotated) pockets were mapped to their respective compound classes using ChEBI ontologies [26]. Only pockets with four or more residues were kept. Inter-chain pockets were split using an all-against-all distance matrix of $C_a$ atoms and distance threshold of 12 Å. This distance matrix was interpreted as an adjacency matrix,

**Table 1. Summary of proteins and pockets for all eleven species used in this study.**

| Species | Abbr. | Kingdom | UniProt acc. | # pred. struct. | # known pockets | # known pockets (exp.) | # pred. pockets |
|---|---|---|---|---|---|---|---|
| *Escherichia coli* | ECOLI | *Bacteria* | 625 | 3,813 | 1,014 | 224 | 2,648 |
| *Saccharomyces cerrevisiae* | YEAST | *Fungi* | 2311 | 5,623 | 1,207 | 109 | 3,000 |
| *Candida albicans* | CANAL | *Fungi* | 559 | 5,622 | 400 | 11 | 3,124 |
| *Arabidopsis thaliana* | ARATH | *Plantae* | 6548 | 24,778 | 4,668 | 169 | 11,075 |
| *Oryza sativa* | ORYSJ | *Plantae* | 59680 | 30,708 | 1,925 | 25 | 10,618 |
| *Zea mays* | MAIZE | *Plantae* | 7305 | 35,165 | 681 | 9 | 13,177 |
| *Glycine max* | SOYBN | *Plantae* | 8827 | 47,539 | 1,220 | 0 | 20,915 |
| *Drosophila melanogaster* | DROME | *Animalia* | 803 | 12,389 | 1,110 | 19 | 5,867 |
| *Caenorhabditis elegans* | CAEEL | *Animalia* | 1940 | 17,819 | 1,091 | 10 | 8,850 |
| *Mus musculus* | MOUSE | *Animalia* | 589 | 20,367 | 4,439 | 146 | 10,133 |
| *Homo sapiens* | HUMAN | *Animalia* | 5640 | 19,268 | 4,451 | 975 | 8,108 |

"# pred. structures" corresponds to the number of protein structures in the AF database without fragments and small proteins. "# known pockets" corresponds to the number of pockets found in UniProt and the "# pred. pockets" to the number of pockets predicted by P2Rank. Note that the pocket sets (known/ predicted) are not mutually exclusive.

and distinct components were determined using scipy [27]. The ChEBI ID for each ligand was used as input for a recursive search on ChEBI ontologies. If a ChEBI ID could not be mapped to any of the compound classes, it was assigned as "other compound". An overview of all compound classes considered in this study is listed in Table 2.

## Compound binding site/ Pocket detection

Potential binding sites were predicted for each investigated protein using P2Rank in conservation mode [6]. P2Rank is a state-of-the-art, machine-learning-based program that was trained to predict potential binding sites via the solvent-accessible surface (SAS) of a protein. After generating a regularly distributed set of points lying on the SAS, feature

**Table 2. Summary table for the compound classes of the ligands binding to known pockets.**

| Compound class | ChEBI ID | # Pockets ("Found") | # Pockets ("Not found") | Odds ratio | $p_{adj}$-value |
|---|---|---|---|---|---|
| Nucleobases, nucleosides & nucleotides | CHEBI:18282, CHEBI:33838, CHEBI:36976 | 8,092 | 2,244 | 8.55 | <<0.001 |
| Inorganic ions | CHEBI:36914 | 1,191 | 7,265 | 0.05 | <<0.001 |
| Other compounds | – | 1,114 | 296 | 3.72 | <<0.001 |
| Carbohydrates & derivatives | CHEBI:16646, CHEBI:63299 | 548 | 103 | 5.08 | <<0.001 |
| Amino acids & derivatives | CHEBI:33709, CHEBI:35238, CHEBI:83821 | 260 | 135 | 1.79 | <<0.001 |
| Lipids | CHEBI:18059 | 173 | 94 | 1.7 | <0.001 |
| Hetero nuclear clusters | CHEBI:33733 | 84 | 433 | 0.17 | <<0.001 |
| Oligo- & polypeptides | CHEBI:25676, CHEBI:15841 | 79 | 92 | 0.79 | 0.387 |
| Monosaccharides & derivatives | CHEBI:35381, CHEBI:63367 | 71 | 34 | 1.92 | 0.006 |
| Glycans | CHEBI:37163, CHEBI:167559 | 73 | 4 | 8.52 | <<0.001 |
| Hemes | CHEBI:30413 | 13 | 2 | 5.97 | 0.003 |
| Nucleic acids | CHEBI:16991, CHEBI:33697 | 0 | 7 | 0.0 | 0.002 |
| No known compounds | – | – | – | – | – |

For all compound classes considered in this study, the respective ChEBI IDs, the number of pockets identified ("found") and not identified ("not found") by P2Rank and the corresponding statistic of the Fisher's exact test adjusted p-value (see Materials and Methods) are listed. "No known compounds" is listed for reasons of completeness of considered categories only.

descriptors are calculated, and a ligandability score is predicted for each of the points. Points with a high ligandability are then clustered, and the respective amino acid residues form the predicted binding sites. These pockets are ranked by the cumulative ligandability scores.

In this study, P2Rank was used with its standard settings, as those do not make use of the Debye-Waller B-factor as a feature. For AF-predicted structures, the predicted local distance difference test (pLDDT) is provided in place of the B-factor, which indicates the confidence of the predicted position for each residue and should not be mistaken as B-factors. Pockets with a mean pLDDT < 70 or a mean predicted aligned error (PAE) > 10 Å were excluded from further analysis, as they are found either in low-quality or intrinsically disordered regions. Both thresholds were set based on visual inspection of the obtained distributions.

The overlap between computationally predicted and the respective known (annotated) pockets from the UniProt database was determined for each protein. If there was any overlap between the amino acid residues of a known and a predicted pocket, the respective known pocket was marked as "found" and the predicted pocket was mapped to the respective compound class corresponding to the ligand that is known to bind to that pocket.

Each compound class was tested for enrichment/ depletion in found pockets compared to all other compound classes using Fisher's exact test. Odds ratios were computed to signify the fold increase or decrease of detecting sites for the respective class from a 2x2 contingency table of observations (found/not-found x is-class/other-class). Fisher exact test p-values for each compound class were corrected for multiple testing using the Benjamini-Yekutieli procedure to control the false discovery rate.

## Pocket comparison and dimensionality reduction

Many different binding site comparison tools have been published in recent years. A subset of them has been compared on the ProSPECCTs datasets in [28] and [29]. In this study, we decided to use two different binding site comparison tools, ProBis [30] and DeeplyTough [19].

Other binding site comparison tools were not suitable for this study for several reasons. Tools, such as SiteMine [29] or IChem [31], depend on internal binding site prediction tools, thus making it difficult to compare their performance to other binding site descriptors for predicted binding sites as the predicted sites potentially differ between different prediction tools. In addition, most of them were implemented to compare one pocket to a set of known pockets and not to perform all-against-all comparison. This results in a high average pairwise runtime, ranging between 8 and 28 s, e.g., for SiteHopper [32] and SMAP [33], making them unsuitable for large-scale comparisons [29]. Lastly, the descriptors for some tools, e.g., KRIPO [34] are based on pharmacophore features that are defined for a protein-ligand complex. This makes them unsuitable for predicted binding sites, where the ligand is not known, which is the case for most pockets.

ProBiS [30] compares protein binding sites by detecting common surface structural patches. The solvent accessible surface atoms within the binding site are converted to overlapping subgraphs, where each vertex represents one functional group with physicochemical properties, such as hydrogen bond donor or hydrogen bond acceptor. The subgraphs are then compared by constructing a modular product graph containing only vertices with matching physicochemical properties. For each product graph, a maximum clique is identified that corresponds to the substructure common to the two compared binding sites. This approach allows ProBiS to detect local structure matches of the two compared binding sites and outputs easily interpretable alignments to the user. In this study, ProBiS was used at the level of within-species pocket comparisons. As the identification of maximum cliques is computationally expensive, ProBiS is not suitable for inter-species comparisons (11 different species) of all predicted binding sites across all species.

To enable a fast comparison of different pockets via a simple Euclidean distance metric and visualizations using established dimensionality reduction methods, DeeplyTough [19] was employed for the large-scale, inter-species comparisons. DeeplyTough consists of a steerable CNN that is trained to separate dissimilar pockets by a certain margin during the training process. In the first step, a 24 x 24 x 24 Å grid is centered on the defined binding site and each binding site

is converted to a 4D tensor with eight channels each one indicating the presence or absence of an atom or pharmaco-phoric features. These 4D tensors serve as input to a steerable CNN with six convolutional layers. To achieve rotational invariance, the authors used data augmentation and introduced an additional stability loss. This loss drives the descriptors of the original binding site and its randomly rotated copy towards being the same. The output of DeeplyTough consists of 128-dimensional embedding vectors that can easily be compared, clustered, and projected to lower numbers of dimensions.

To plot the resulting 128-dimensional embedding vectors in 2D, different dimensionality reduction techniques were employed. These methods include principal component analysis (PCA), independent component analysis (ICA), uniform manifold approximation (UMAP) and t-distributed stochastic neighbor embedding (tSNE). While for PCA, ICA, and tSNE the implementation in the scikit-learn Python package was used [35], the Python package umap-learn was used for UMAP [36]. For UMAP, n_neighbors was set to 10, 20, 50, 100, and 200 to focus on local neighborhoods (smaller values for n_neighbors) and more global relationships (larger values of n_neighbors) and min_dist was set to 0 to enable clustering. For tSNE, the following parameters were used: n_iter=5000, random_state=1, learning_rate="auto", init="random". Perplexity was set to 10, 20, 30, 40, 50, and 100 to test the influence of local and global shapes within the dataset, and set to 50 for the main results set.

## Calculation of pocket descriptors

As DeeplyTough does not provide any explicitly interpretable features, several simple pocket descriptors (i.e., properties) were calculated for each pocket to identify the most important features that DeeplyTough is learning. These descriptors include physicochemical properties, such as aromaticity, hydrophobicity, net charge, solvent accessible surface area (SASA), and the structure prediction quality parameters: predicted aligned error (PAE) and predicted local distance difference test (pLDDT).

The aromaticity and hydrophobicity of each predicted pocket was calculated using the ProteinAnalysis module of biopython [37]. This module calculates the aromaticity and hydrophobicity of an amino acid sequence according to [38] and [39], respectively. The net charge of each pocket was estimated using the amino acid residues lining each pocket and the charge of the respective residue type at pH 7.0. All neutral residue types were assigned a charge of 0, and all negative residue types (Glu, Asp) were assigned a charge of -1. All positively charged residue types (Arg, Lys) were assigned a charge of +1. Histidine was assigned a charge of +0.5.

The SASA of each pocket was calculated using the Shrake and Rupley algorithm implemented in biopython [37,40]. The sum of all SASA values for each residue in a pocket was obtained and then divided by the maximal solvent accessible surface area for the respective amino acid residue type as defined by [41] to obtain the relative SASA for each pocket.

To check how confident AF is about the relative and local positions of all residues within one binding site, we calculated the mean PAE and the mean pLDDT for each pocket. The PAE and pLDDT values for each protein of interest were obtained from the AF database, with the pLDDT values extracted directly from the corresponding protein structure files [20].

## Clustering and analysis

As DeeplyTough's embedding space for all binding sites was found to be too uniformly populated to obtain a meaningful clustering, only ProBiS alignment scores were used to estimate the number of unique pockets per species, as ProBiS operates at an increased level of structural resolution compared to DeeplyTough. The pairwise binding site score matrix for each species was clustered in two ways. In a simple approach, the similarity matrix for each species was clustered by treating it as an adjacency matrix and applying different thresholds for setting edges. Clusters were then detected by determining the number of components detected per graph. In addition, using the igraph package (R 4.1.2), pocket clusters were created for both weighted, undirected graphs based on the computed ProBiS similariy scores (normalized by the maximum thereof obtained in a given species), and unweighted, undirected graphs by setting an edge between pockets

(nodes) if the max-normalized ProBiS score was greater than 0.1 (determined based on reviewing the score distributions across species), and now edge otherwise. The latter allowed for randomizing the graph under preservation of the degree distribution using the igraph "rewire" function with "keeping_degseq" set active (niter = 10). Furthermore, completely random, unweighted, undirected networks (same number of nodes and edges) were also created and assessed. Clusterings were obtained using the "cluster_leiden" function with the resolution_parameter set to 0.01 (different thresholds were also tested, but were determined not to qualitatively affect the results. Respective results are provided on the github page (see below).

When performing a cross-species analysis of binding site similarity, to estimate the kingdom or species diversity of binding sites mapping to same region in tSNE projection plots, we computed the entropy, $S$, on binned tSNE regions, with

$$S = -\sum_{i=1}^{N} p_i \, log\,(p_i)\,,$$

(Eq 1)

where $p_i$ is the fractional frequency of pockets associated with a given kingdom or species relative to all pockets mapping into the same region (bin) of tSNE space, and $N$ is the number of considered kingdoms or species, respectively.

### Visualizations

All protein structure visualizations were created using ChimeraX v. 1.7 [42]. All other plots were created with the matplotlib [43] and seaborn library [44]. The Plotly library [45] was used to create interactive plots of the embedding vectors produced by DeeplyTough.

## Results

### Pocket identification with P2Rank

A set of 223,091 predicted protein structures found in the AF database for eleven species, including bacteria, plants, unicellular fungi, an insect, and two mammal species (Table 1), was subjected to binding site predictions using P2Rank. Testing various thresholds, we determined the recall, i.e., the detection of known binding sites, was relatively stable across the P2Rank score threshold range from 0.2-0.7, with 59% and 45%, respectively. To balance the number of true positive and potential false positive binding sites, i.e., sites that are not actually involved in any ligand binding, and to arrive at a manageable set size (176,156 pockets for a threshold of 0.2), a threshold of 0.5 was applied to the probability score as predicted by P2Rank for all pockets. This resulted in a set of 97,515 potential binding sites for the eleven different species used in this study. A summary of the known and predicted binding sites of each species is provided in Table 1. The number of predicted pockets is strongly positively correlated with the number of predicted protein structures (Pearson correlation coefficient, r = 0.98). The largest number of pockets was predicted for SOYBN, which also has the largest proteome. The smallest number of pockets is predicted for ECOLI with only 3,809 predicted protein structures. Note that at this stage, no protein or pocket clustering was performed. Thus, this correlation pertains to the raw counts.

First, we wished to verify that the quality of the protein structures is comparable across the species used in this study. The distribution of PAE (AF-predicted aligned error) values associated with all pockets of the 11 species and the applied threshold value of 10 Å is visualized in Fig 1A. The median PAE ranges between 2.9 and 4.3 Å for all species. The median pLDDT ranges between 87.4 and 92.7 for all species. As expected, pLDDT and PAE medians show a negative correlation. High pLDDT values (ECOLI) are matched by low PAE, whereas low pLDDT (ORYSJ) are mirrored by a high PAE median. The largest variation in both the PAE- and the pLDDT-values was observed for MAIZE, ORYSJ, and SOYBN. These species have the largest proteomes and the most predicted pockets. The least variation was observed for ECOLI, suggesting a lower fraction of less-defined, disordered regions in ECOLI. Notwithstanding the species-related differences, we considered all AF-predicted protein structure sets suitable for a systematic cross-species comparison. Based upon

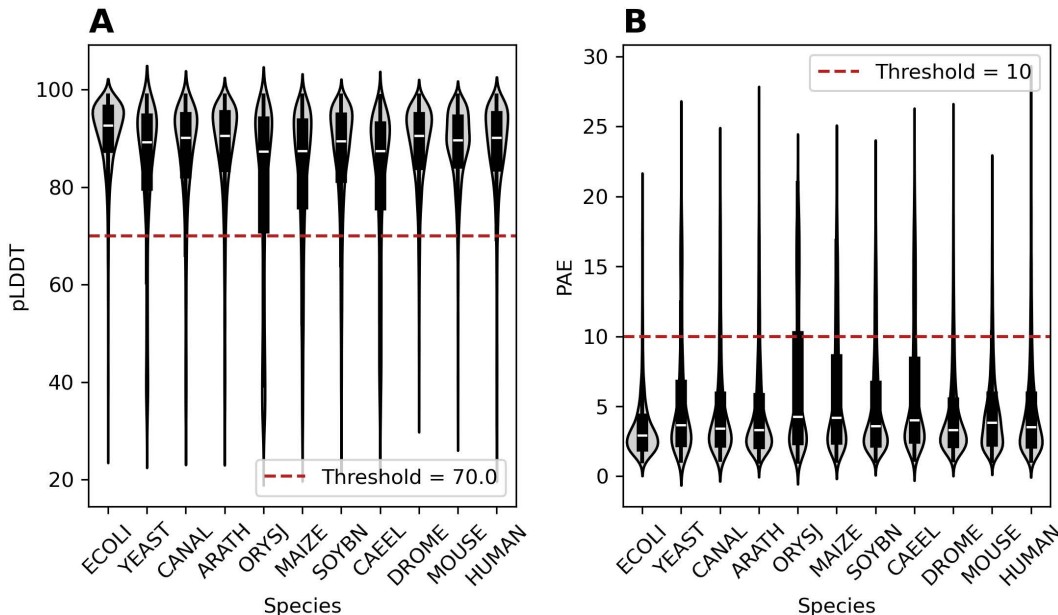

**Fig 1. Structural confidence of pocket conformations as judged by AF-Predicted Aligned Error (PAE) and predicted local distance difference test (pLDDT).** (A) Violin plots for PAE values per-species. PAE-values correspond to the average of the pairwise all-against-all residue PAE matrix. All pockets above the red dotted line were not considered for further analysis. The median for all PAE distributions varies between 2.9 Å and 4.3 Å. MAIZE shows the largest variation in PAE values. (B) Violin plots for pLDDT values per-species. pLDDT-values correspond to the average of the respective pLDDT-values found for all residues within a pocket. All pockets below the red dotted line were not considered for further analysis. The median for all pLDDT distributions varies between 87.4 and 92.7. MAIZE shows the largest variation in pLDDT values as well.

visual inspection of the PAE and pLDDT distribution plots, we set the thresholds for including pockets to PAE < 10 Å and pLDDT > 70. Of note, across all 11 species, pLDDT values were generally higher in proteins with detected binding sites than in proteins without binding sites (S1 Fig), likely explained by low pLDDT scores reflecting disordered regions or proteins.

If available for a given protein, the domain information was mapped onto the respective predicted binding pockets. While 34,101 (35%) pockets could not be assigned to any domain, most of the pockets that were assigned to a single domain are annotated as protein kinase domains (InterPro: IPR000719, 4,007 pockets) and GPCR, rhodopsin-like, 7TM domain (InterPro: IPR017452, 2,083 pockets). A total of 38,929 pockets predicted by P2Rank for all species are inter-domain binding sites that are assigned to two or more different domain types. Most of these pockets are at the interface between two different domains, but 9,979 pockets are found at the interface of more than two domains. Another subset includes pockets that were found at a "domain-no_domain" interface (9,979 pockets). A small subset of 583 pockets was found in domains of unknown function (DUF).

Fig 2 shows the number of residues per binding site, i.e., residues lining the respective pocket reflecting on binding site size, for known and predicted pockets. Predicted pockets are, on average, significantly larger (median number of residues = 24) than known binding sites (median = 6). While known pockets are derived from experimentally solved structures with oftentimes manually annotated binding site residues, P2Rank predicts binding sites by clustering potentially druggable surface atoms. Therefore, residues not directly involved in the binding of any ligand are potentially included in the definition of predicted pockets as well. In addition, P2Rank does not include any additional pocket segmentation, resulting in large pockets if several ligands are binding to a similar region, as is, for example, the case for ATP-dependent reactions.

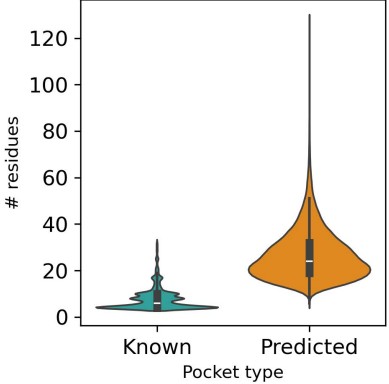

**Fig 2. Pocket size statistic as gauged by the number of residues lining a given pocket in known and predicted pockets.** On average, predicted pockets consist of more residues than experimentally annotated (known) pockets. While the median number of residues for known pockets is 6, the median is 24 for predicted pockets.

Exploring the reliability of P2Rank pocket predictions by checking the overlap of predicted sites with known sites, designating them as "found" if at least one residue-based overlap was detected, and "not found" otherwise, we observed that the most "found" as well as "not found" pockets were annotated as binding "nucleobases/ nucleosides/ nucleotides". An 8.55-fold enrichment (p-value << 0.05, Fisher's exact test) for known pockets that are denoted as "found" was observed for this ligand class as compared to all other ligand classes. About 15% of the "nucleobases/ nucleosides/ nucleotides" pockets that were not identified were inter-chain pockets. For inorganic ions, the second largest compound class (Fig 3), we observed a significant depletion with p-value << 0.05 and odds ratio = 0.05 (Fisher's exact test) for known pockets that are denoted as "found" as compared to all other ligand classes. The binding sites of inorganic ions are often formed by a few amino acid residues, via charge-driven complex formation or covalently, thereby exhibiting less of the hallmark features of compound binding sites (surface invaginations). This makes it difficult to predict their location if they are bound to a site that is distant from any other binding site. Thus, inorganic ions are proportionally overrepresented in the "not

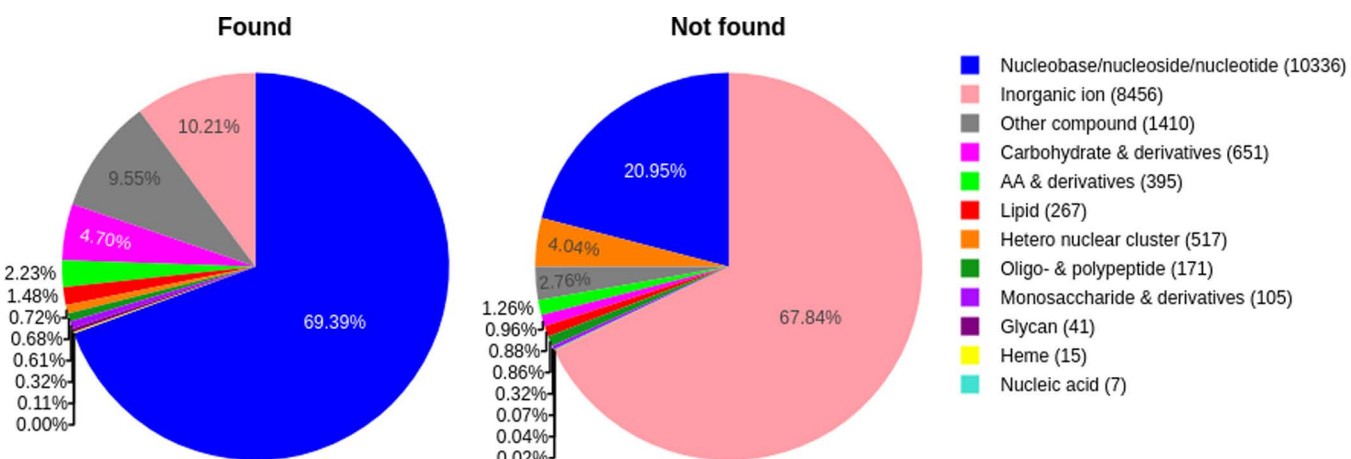

**Fig 3. Ligand annotation of pockets.** Known (i.e., annotated) pockets identified by P2Rank used as the binding site predictor and with correspondingly available ligand annotation were classified as "found" (n = 11,662, 52.1%), and "not found" (n = 10,709, 47.9%), otherwise. The total number of occurrences for each compound class is denoted in brackets in the legend.

found" set. For other ligand classes, there are significantly fewer binding sites annotated in UniProt. These classes include heme, lipids, carbohydrates and derivatives, and amino acids and derivatives. A significant enrichment for pockets that were "found" was observed for "Glycans", "Carbohydrates & derivatives", "Amino acids & derivatives", "Lipids" and "Mono-saccharides & derivatives" while a significant depletion of pockets that were "found" was observed for "Hetero nuclear clusters". For "Hemes", all pockets were found. For nucleic acids, none of the pockets could be identified by P2Rank using a threshold of 0.5 for the probability. "Oligo- and polypeptides" were found at the expected average rate, odds-ratio = 0.79, p = 0.387. A summary of the respective statistics of the Fisher's exact test and the number of pockets denoted as "found" and "not found" for all ligand classes can be found in Table 2.

Several factors may explain why known pockets might not be predicted by P2Rank. One reason is that flexible loops flanking the binding site lack a stable conformation. These flexible loops are often missing in experimental structures. In AF, they are assigned to a single conformation with a mostly low pLDDT values. An example for this case is shown in Fig 4A. While the two flexible regions are missing completely in the experimental structure of tubulin polyglutamylase TTLL7 (PDB ID: 4YLR), the ADP binding site is not accessible in the AF predicted structure (UniProt ID: Q6ZT98). Other known binding sites, such as the LacNAc binding site on galectin-3, only consist of few residues that form a flat binding surface (see Fig 4B).

### Per-species binding site comparison (ProBiS)

As we wished to discern the underlying scaling law linking the pocketomes of different species to their respective pro-teomes viewed from an evolutionary perspective, we performed a per-species comparison of all predicted binding sites using ProBiS to determine the number of unique binding sites present in each species. This graph-based comparison algorithm results in a sparse similarity matrix, where zeros denote pocket pairs that cannot be aligned and thus are considered dissimilar. The higher the alignment scores, the more similar two pockets are. Pockets that could not be aligned to any other pocket of the respective species were termed "singletons". The number of singletons and the number of communities as detected by the Leiden algorithm are listed in Table 3. The fraction of "number of singletons" over the "number of predicted" pockets is smaller for species with larger proteomes and more predicted pockets than for species with smaller proteomes, such as ECOLI, YEAST, and CANAL. For all three species, about 40% of all pockets were singletons. The smallest fraction of singletons over the number of pockets (0.079) was detected for SOYBN. The inverse relationship (Pearson correlation r = -0.84, p = 0.001) between proteome size and fraction of singletons is, to some degree, to be expected, given that the pocket space crowds up more with more proteins and their associated pockets. It would, however, be different if more proteins meant altogether new and distinct pockets, a question we will address to below.

As singletons might potentially be falsely predicted binding sites that are not actually involved in any ligand binding, as it may seem unlikely that proteomes harbor sites that occur only once and thus for a single-case ligand binding event, we

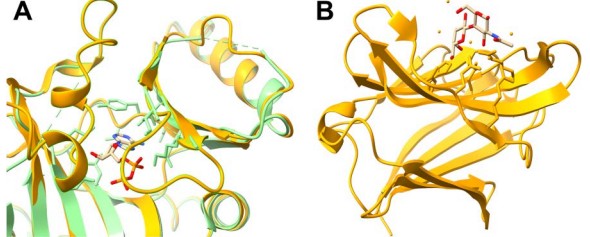

**Fig 4. Examples of binding sites that could not be identified by P2Rank due to flexible regions and flat surfaces.** (A) Experimental (PDB ID: 4YLR, orange) and AF structure for Q6ZT98. In 4YLR, ADP (beige) is bound as a ligand. This ADP binding site is blocked in the AF structure (green) by two flexible regions (residues: 161–172, 235–269) that are not present in the experimental structure. Therefore, P2Rank cannot predict the ADP binding sites. (B) Galectin-3 (PDB ID: 1KJL, orange) binds LacNAc (beige). The binding site is found on a flat surface and consists of only few AAs.

**Table 3. Pocket clustering statistics.**

| Species | # Communities | # Singletons | # Singletons/ # Pockets | Largest community |
|---|---|---|---|---|
| ECOLI | 273 | 1,076 | 0.41 | 185 |
| YEAST | 335 | 1,182 | 0.39 | 133 |
| CANAL | 300 | 1,293 | 0.41 | 124 |
| ARATH | 823 | 1,532 | 0.14 | 893 |
| ORYSJ | 780 | 1,711 | 0.16 | 924 |
| MAIZE | 966 | 1,879 | 0.14 | 1,086 |
| SOYBN | 1,376 | 1,792 | 0.09 | 1,868 |
| DROME | 514 | 1,688 | 0.29 | 328 |
| CAEEL | 617 | 2,225 | 0.25 | 406 |
| MOUSE | 708 | 1,805 | 0.18 | 1,068 |
| HUMAN | 682 | 1,705 | 0.21 | 516 |

Communities, i.e., sets of similar pockets, were found by applying the Leiden algorithm on unweighted graphs (binary similarity matrix based on threshold-based discretization of the ProBiS scores, see Materials and Methods). Singletons are lone pockets, not similar to any other pocket in a given species. While the number of singletons is getting larger, the number of singletons per pocket is getting lower with larger proteomes. The highest fraction is observed for CANAL, ECOLI, and YEAST. These species have the lowest number of communities. The lowest fraction is observed for ARATH and SOYBN. "Largest community" indicates the number of member pockets in the largest found community.

compared various properties of the pockets for singletons and non-singletons, such as P2Rank probability, PAE, pLDDT, number of residues, solvent-accessible surface area (SASA), aromaticity, net-charge, and hydrophobicity Fig 5). All of the tested properties show significant differences between singletons and non-singletons with p-value < 0.01 (t-test), but the actual effect sizes, calculated using Cohen's d, differ markedly across the various properties. The largest effect size was observed for the P2Rank probability of the pockets, i.e., the confidence of the pocket prediction. The median probability of the predicted pockets drops from 0.83 for the non-singletons to 0.67 for singletons with an associated effect size of 0.738 (Fig 5G). The effect size for all other properties were found in the range 0.053 to 0.468 (see Fig 5). Taken together, singletons differ markedly from non-singleton pockets, which may indicate that they are either special because of their respective particular binding modes and ligands outside the P2Rank training sets and score-tuning approaches, or that they may be falsely predicted pockets.

Next, we addressed the question as to whether expanding proteomes during the course of evolution are commensurate with proportionally increased pocketome sizes. While a proportional increase may be the naive expectation, alternatively, relatively more or fewer pockets may have emerged during evolution than is expected purely based on proportionality. That is, we wished to understand whether evolution "invests" in new proteins to expand the set of diverse compounds that can be processed (via binding to proteins), or whether functions other than compound binding are invested in. In case of the latter, with more unique protein folds, fewer than proportionally more binding pockets would be found with increased proteome sizes. By contrast, evolution could also proceed "economically" by reusing the same fold but evolving new and additional binding sites on them via selected point mutations. Under that scenario, more than proportionally more pockets per protein fold would be seen.

To explore this relation between unique binding sites and unique folds across species, we plotted the number of unique binding sites (communities + singletons) against the number of FoldSeek (FS) clusters (https://cluster.foldseek.com/) [25] and the number of clusters (communities only) against the number of FS clusters. We decided to gauge the proteome size as the number of FS clusters and not by the number of encoded proteins to avoid wrongly associating different sizes of protein families (paralogs) with protein "novelty". This approach also accounts for the effect of gene duplication in some plants, such as SOYBN, where 73.6% of the genes are duplicated according to its BUSCO score. Note that unlike reported above, where we stated a high linear correlation of numbers of proteins and pockets, here we performed a clustering of both entities to really capture unique proteins and unique pockets.

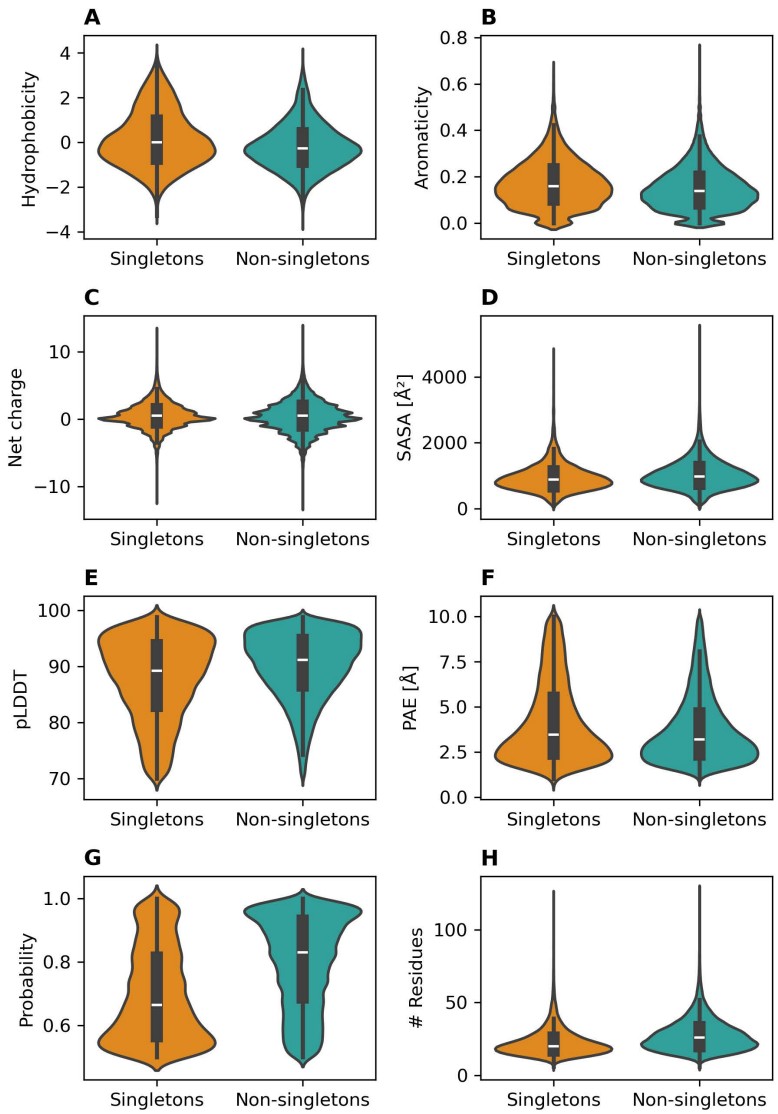

**Fig 5. Annotation confidence of pockets for non-singletons and singletons.** Violin plots of various properties contrasted for singletons and non-singletons (A) hydrophobicity, effect size (Cohen's d) between the two groups is 0.334. (B) aromaticity, d = 0.207. (C) net charge d = 0.053. (D) solvent accessible surface area (SASA), d = 0.230. (E) pLDDT, d = 0.369. (F) PAE, d = 0.223. (G) P2rank, d = 0.738. (H) number of residues, d = 0.468.

As shown in Fig 6A, there is an inverse association between the number of FoldSeek clusters, $N_{FS}$, in the different species and the number of pockets clusters per FS cluster, $N_{Pocket\ clusters}/N_{FS}$, where $N_{Pocket\ clusters}$ is the number of pocket clusters in a given species. Compared to ORYSJ, which has the lowest ratio of pocket clusters to FS clusters, the ratio between the number of pocket clusters and FS clusters is about four times higher for ECOLI, which has only about 1/7th the number of FS clusters compared to ORYSJ. Note that under pure proportionality, we would expect this relationship to be a constant (as we normalize the number of pocket clusters by the number of FS clusters for better visual clarity). Regressing $\log(N_{Pocket\ clusters})$ as a linear function of $\log(N_{FS})$, a strong linear relationship (Pearson correlation coefficient $r = 0.942$) was obtained, suggesting a power law relationship with proportionality of $N_p \sim N_{FS}^a$, and $a = 0.398$ (slope of the regression line, unweighted graph), i.e., substantially smaller than one. A smaller than one exponent $a$ suggests a less than linear

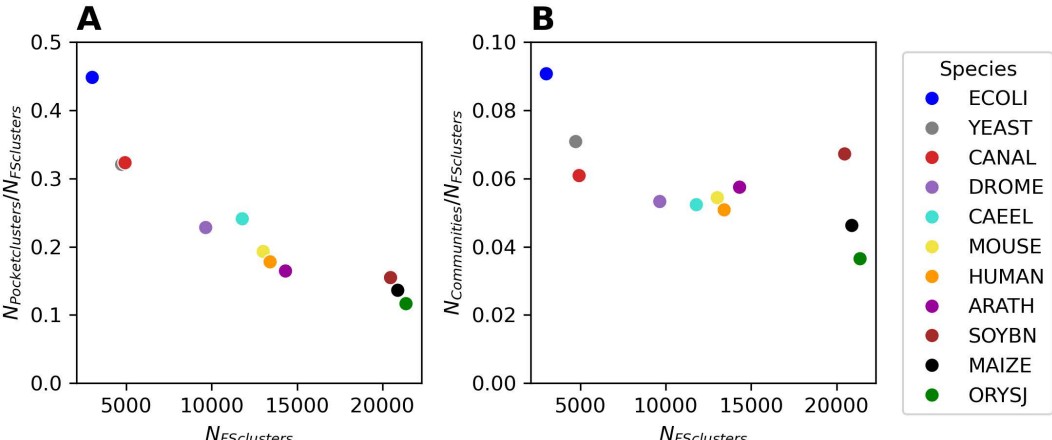

**Fig 6. Scaling law of the number of unique pockets vs. proteome size.** (A) Number of unique pockets per species ($N_{Pocket\ clusters}$) normalized by the number of FoldSeek (FS) clusters vs. the number of FS clusters ($N_{FS\ clusters}$). On a log-log scale for $\log(N_{Pocket\ clusters}) \sim \log(N_{FS\ clusters})$, a linear regression yielded: slope = 0.398, intercept = 4.012, Pearson r = 0.942. (B) Number of pocket communities per species, $N_{Communities}$, i.e., without considering singleton pockets, normalized by the number of FoldSeek (FS) clusters vs. the number of FS clusters ($N_{FS\ clusters}$). On a log-log scale for $\log(N_{Communities}) \sim \log(N_{FS\ clusters})$, a linear regression yielded: slope = 0.73, intercept = -0.364, Pearson r = 0.952). Note that the reported slope values correspond to the power law exponent, *a*, discussed in the main text. The reported results relate to the unweighted (binarized) network (see Materials and Methods).

(sub-linear) increase of the number of unique pocket clusters with increasing number of FS clusters, i.e., larger and more divers proteomes do not harbor a proportionally greater diversity of binding sites. For the original ProBiS-score-weighted graph, *a* = 0.44 was obtained.

To obtain a null-model, we randomized the unweighted graphs (binary adjacency matrix), preserving the degree distribution (the actual network was found to follow a power-law degree distribution (see below)). From five performed randomization runs, an average scaling law coefficient, *a* = 0.882 +/- 0.001 (s.d.), for the randomized network was obtained. i.e., substantially larger than the one obtained for the actual network (*a* = 0.398). Thus, compared to random expectation, evolution shows evidence of a pocketome diversity plateau. Despite more and different protein structures appearing, not proportionally and less than randomly expected different binding sites appeared in evolution. Note, under a completely random graph model (randomizations performed without degree-preservation, but preserving node and edge counts only, randomly assigned), we obtained a very loosely defined (r = -0.42 +/- 0.01 (s.d.) negative coefficient (*a* = -0.98 +/- 0.01 (s.d.)), which, however, is a) dictated by outliers (mouse clusters into one cluster), and b) is not the proper reference network, as the actual network is a power-law node degree network. Furthermore, for the original ProBiS-score-weighted network, degree-preserving randomizations are not possible; they require unweighted graphs (binary adjacency matrix). For completely randomized ProBiS-score input matrices (kept symmetric), *a* = 0.45 +/- 0.001 (s.d.) was obtained.

As reported above, singleton pockets, i.e., pockets not found similar to other pockets within a given species, differ markedly from non-singletons with regard to their properties. To assess the effect of singletons, which may be false-positive pockets, on the scaling law statistic, we performed analyses ignoring singletons. Excluding singletons, we determined a power law exponent of *a* = 0.73 for the actual network, and *a* = 0.925 +/- 0.002 for degree-preserved random networks. Thus, even without considering singletons, actual pocket communities grow more slowly than randomized versions and are distinctly lower than expected for proportionality, albeit the difference is smaller for communities compared to the overall clustering statistic, i.e., including singletons.

Regarding the sizes of the communities, the Leiden algorithm detects a few large communities for each species and many smaller communities that only consist of two or three different pockets (Fig 7). Overall, linear relationships are observed in the log-log plot of cluster-frequency and cluster size, suggesting an overall power-law relationship for the

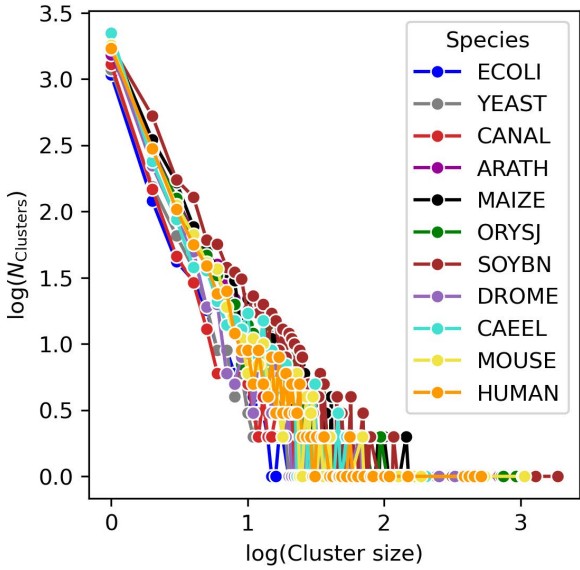

**Fig 7. Cluster size frequency distribution for the eleven species used in this study.** The log(Cluster size) plotted against the log(NClusters). The clustering for all species reveals that there are many clusters that only contain few pockets and only few clusters with many pockets, and overall, a power-law degree distribution is observed. The largest clusters are found in SOYBN, MAIZE, MOUSE.

cluster-size distribution, which is oftentimes observed for such data types in biological systems [46]. The largest communities are found in SOYBN and MAIZE, whereas in ECOLI, the distribution is shifted to smaller clusters. Thus, as expected, the size distribution of clusters correlates with set size (number of pockets, Table 1).

As ATP is the most frequent known ligand (n = 6,327 for ChEBI:30616) and protein kinase is the most frequent domain annotated for the predicted pockets, it is expected that the respective largest communities are enriched for protein kinase domain annotations. This is the case for ARATH, ORYSJ, MAIZE, SOYBN, and CAEEL. For these species, 81.03% (CAEEL) to 96.41% (SOYBN) of the pockets in the largest cluster are found at least partially in a protein kinase domain. For YEAST, CANAL, DROME, and HUMAN, the protein kinase domain dominates in the second largest cluster with 127, 105, 253, and 457 members, respectively. The two largest communities in MOUSE contain mainly pockets that are found at least partially in the "GPCR, rhodopsin-like, 7TM" domain. The fourth largest community in MOUSE is enriched for pockets found in the "Protein kinase" domain. The paucity of protein kinase annotations in ECOLI is expected, as it does not have the eukaryotic protein kinase domain. The largest proportion of pockets is found at least partially in the "ABC transporter-like, ATP-binding" domain (74 of 185 pockets in the largest community) for ECOLI.

**Large-scale, cross-species binding site comparison (DeeplyTough)**

Thus far, we clustered pockets and proteins per species. The pocket datasets could also be inspected across all species. As the computational resources needed for an all-against-all comparison using ProBiS are prohibitively high, we decided to use the embedding vectors obtained by DeeplyTough for a large-scale, inter-species comparison. By using DeeplyTough, a method that describes pockets as numeric vectors of length 128, we were able to perform an all-against-all pocket comparison of all 97,515 pockets across all eleven species used in this study, arriving at a global view of the pocketome "universe". We present this view primarily by using tSNE (perplexity = 50 (~number of neighbors considered during embedding), tSNE maps using different settings are provided in S2 – S6 Figs) as an appropriate projection method, but included projections for other perplexity values, UMAPs with varying number of neighbors, PCA (principal component analysis), and ICA (independent component analysis) as well as part of supporting material (see S7 – S13 Figs).

Fig 8 shows all pockets projected into the first two tSNE components colored by the pocket properties: hydrophobicity, aromaticity, net charge, solvent accessible surface area (SASA), pLDDT. PAE, number of residues, and probability. Each point in the plot represents one unique pocket, allowing us to gauge the most important properties by which the pockets segregate. Most strongly, pockets differ by hydrophobicity, to a lesser degree by aromaticity, charge, and the quality parameters pLDDT, PAE, and pocket probability. Size-related properties (SASA and number of residues) do not cause a notable segregation of pockets.

To investigate whether particular compound classes correspond to binding sites displaying characteristic properties, we examined the pocketome projection with regard to their annotation, i.e., which compounds are known to bind to it, provided this information is available (Fig 9). The most abundant ligands found in known pockets are "nucleobases, nucleosides, and nucleotides", such as ADP and ATP. The predicted pockets that overlap with these pockets are mostly found on the right side of the tSNE plot. As evident from Fig 8, these pockets are hydrophilic to neutral on the hydrophobicity scale and are predicted with moderate to high probability. In addition, they contain few aromatic amino acid residues. As

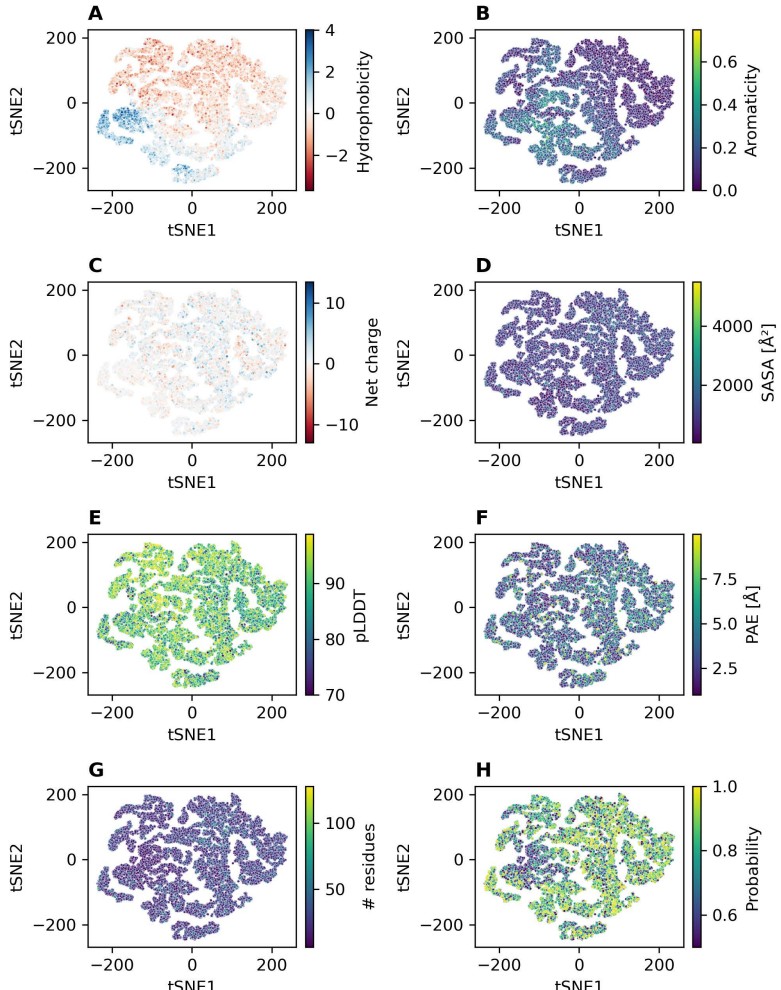

**Fig 8. tSNE plots for all 97,515 predicted pockets of all eleven species colored by different properties.** Perplexity set to 50. tSNE plots obtained for other perplexity settings are provided in the supporting information S2–S6 Figs.

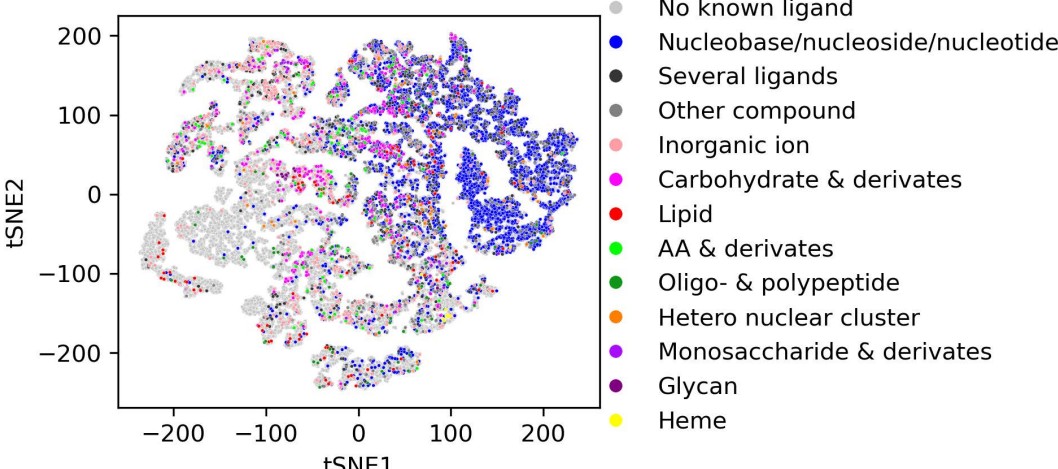

**Fig 9. tSNE plot of all pockets of all eleven species colored by different ligand classes.** The most abundant ligand class is "nucleobases/nucleosides/nucleotides", which are mostly found on the right side of the tSNE plot. While glycan binding sites cluster well, the binding sites for other ligand classes are found in different regions of the tSNE plot. An interactive plot for further inspection is made available as well (see Data availability).

mentioned above, P2Rank does not perform any sub-sampling of the predicted pockets. Binding sites that contain multiple ligands are therefore predicted as one large pocket instead of several separate pockets.

Glycan binding sites are rather hydrophilic, contain moderate amounts of aromatic amino acid residues, and are all found in the Glycoside hydrolase family 16 domain (InterPro: IPR000757). They are found on the left side of the tSNE plot and cluster well. One reason for that might be their unique shape as they are formed by a set of anti-parallel β-sheets and their connecting loops (example shown in Fig 10). In the vicinity of these pockets, there are several other, not annotated pockets that are found on the glycoside hydrolase family 16 domain. This suggests that more glycan binding sites could be potentially identified with this approach.

For other ligands, such as lipids, carbohydrates, and amino acid derivatives, the predicted binding sites that overlap with known binding sites of those compounds do not cluster strongly. Their associated binding sites are found in several parts of the tSNE plot.

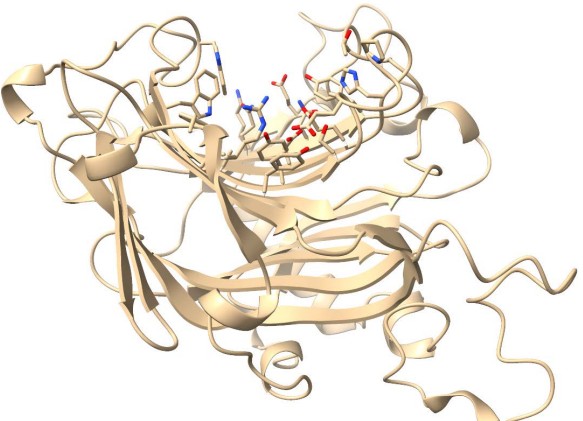

**Fig 10. Glycan binding site of the protein Q8LC45.** The binding site residues are displayed as sticks, other parts of the protein as cartoon. The binding site is formed by several anti-parallel β-sheets. This protein visualization was created using ChimeraX (v. 1.7).

On the left side of the tSNE map, there are two regions where only few pockets could be annotated. The first region, marked in orange in S14 Fig, contains about 12,000 pockets. The median probability for these pockets is significantly lower than for the rest of the pockets (0.65 vs. 0.83), and about one-third of these pockets are at least partially found in a transmembrane domain. Regarding the domains to which these pockets are assigned, about 47% of these pockets could not be assigned to any domain. The rest of the pockets are assigned to various other domains with "GPCR, rhodopsin-like, 7TM" domain being the most common domain. The second region corresponds to the cluster to the left in the tSNE map. This region contains about 3,650, mainly hydrophobic pockets (median hydrophobicity = 1.858) that were predicted by P2Rank with a slightly higher median probability than the rest of the pockets (0.842 vs. 0.806). As noted for the first described region, about half of the pockets found in this region could not be assigned to any domain, while the other pockets are assigned to various other domains.

As we wished to discern kingdom- and species-specific differences between the pocketomes, we colored the tSNE projection by kingdom of life and species. The kingdom-specific projection map shown in Fig 11 does not reveal any strong kingdom-specific differences for *bacteria*, *fungi*, *plantae*, and *animalia*, with pockets from all kingdoms occupying all of the tSNE space (Fig 11A) and the binned entropy heatmap capturing diversity revealing a rather uniform kingdom representation. The only region that contains fewer binding sites found on bacterial proteins, is the upper right part of the tSNE plot (Fig 11A and 11B). This corresponds to the region where most of the pockets are found in the protein kinase domain. This eukaryotic domain is not found in any bacterial protein. Similarly, no species-specific binding sites are revealed in the respective tSNE plot shown in S15 Fig.

Taken together, these results suggest that either compound binding is not very specifically associated with pocket descriptors, or, and we consider this to be more likely, the binding site definition of DeeplyTough's embeddings may be too coarse-grained to capture local similarities between different binding sites or differences, respectively. The binding sites are rather separated by their overall physicochemical features and high-level domain information. This leads to an embedding space that is populated too uniformly to allow for meaningful fine-grained clustering.

## Discussion

In this study, we aimed at generating a global view of the pocketome universe. Inspecting the eleven species for which the AlphaFold database contains a comprehensive set of predicted protein structures and applying established binding site

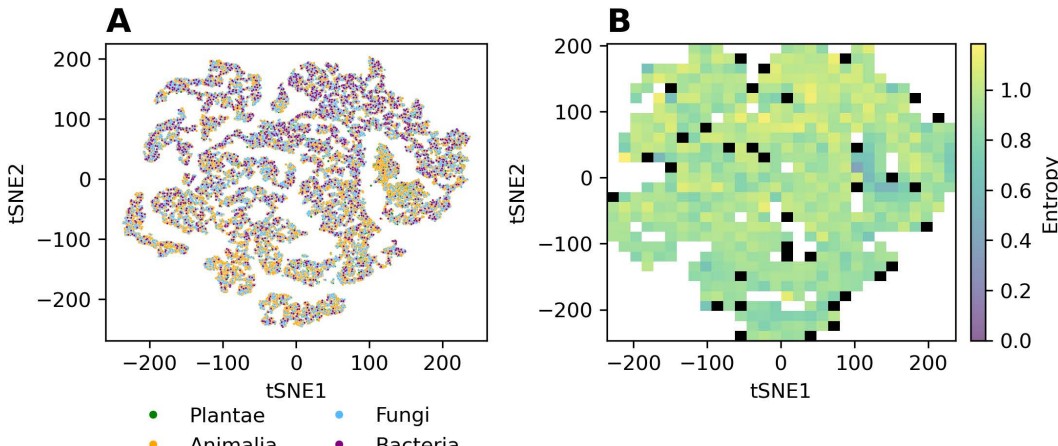

**Fig 11. tSNE 2D-projection plots colored by the respective kingdoms of life.** (A) All pockets in their respective tSNE embedding space, (B) Binned entropy, S (see Materials and Methods), heatmap plot, capturing for every square bin the kingdom diversity with low S values indicating low diversity and high values corresponding to respectively increased diversity. All grid cells containing fewer than 20 pockets are colored black, as no robust entropy estimate can be computed. An interactive plot (also for species-level annotations) for further inspection is made available as well (see Data availability).

(pocket) prediction algorithms, we performed a systematic analysis of binding site properties, ligand sets, clustering, and their scaling with proteome sizes.

Our analysis suggests that during the course of evolution, with species evolving larger proteomes and more differentiated molecular functions, unlike perhaps expected, the increase in proteome size and complexity is not commensurate with a proportional increase in binding site "innovation". Instead, fewer than proportionally expected unique binding sites seem to have emerged in evolution (Fig 6A). This implies that, rather than compound binding sites, other functions may have emerged and expanded, such as possibly transcriptional and translational regulation and signaling. However, it needs to be cautioned that this conclusion is based on the approach pursued here, with all the caveats pertaining to the used tools and settings (detailed below).

How evolutionary increases in molecular repertoire (genome-size, number of genes etc.) relate to increases in functional diversity has been studied before, in particular in recent years, with close-to-comprehensive coverage of sequence-and-structure space, whether there is evidence of a saturation of sequence-structure-function space exploration by evolution [47,48]. Notably, Nimwegen [47] reported on scaling laws relating number of genes with the size of various functional categories (number of genes devoted to the respective functions) based on gene-ontology, found for all but one sub-linear scaling laws, with the notable exception of "signal transduction", for which higher than linear (exponent 1.55-1.90) were reported, in line with our hypothesis stated in the previous paragraph. With regard to structural space and its association with function, it was repeatedly reported that both same-structure-different-function and different-structure-same-function can be observed [49,50]. Our results suggest that at large scale, across kingdoms, the functional space (with pocket diversity taken as a proxy of functional capabilities) seems more constrained than structure space and that the predominating mode of evolution is not the invention of novel function using the same fold, as this would have resulted in a greater than linear scaling law, but rather that functional novelty requires disproportionally more novel structures. Interestingly, a recent survey of microbial protein structures and exploiting the recent leap in structure information coverage suggested that structure space is already largely saturated [48] which, in light of our findings, may mean that functional diversity is also saturated, and has been saturated already earlier (in evolution) than the structural space.

While the pocketome universe exhibits a partitioning into pockets of particular physicochemical properties (Fig 8) and compound class binding (Fig 9), we did not identify any such segregation with regard to species or kingdom (Fig 11 A and 11B), except for the missing protein kinase domain in bacteria. This may either result from technical limitations to discern any present species/kingdom-specific pockets (discussed below), or may suggest that species/kingdoms are not characterized by specific binding sites. However, as, for example, plants do contain specialized metabolic reactions (secondary metabolism), it seems surprising that this should be the case. Yet, as those specialized compounds are consumed and degraded by other species as food, it may be a wrong assumption to expect unique pockets in plants, as in species that feed on plants, similar binding sites need to exist to process the compounds.

One source of uncertainty possibly affecting our analyses concerns the reliability of predicted pockets. As P2Rank [6] does not perform any sub-sampling, the predicted pockets are, on average, larger than known pockets (Fig 2) and are likely to merge several distinct pockets, i.e., for pockets that are found in protein kinase domains. Introducing a sub-sampling algorithm as a post-processing step of P2Rank could improve the precision of the binding site comparison as well as the quality of the P2Rank predictions in general.

One option to potentially refine predicted binding sites may be to exploit domain information. In our study we identified a subset of 32,098 pockets that are found at a domain - no_domain interface, i.e., sites for which some residues belong to an annotated domain, while others carry no domain annotation information. Those pockets may be less well defined structurally than pockets fully embedded in specific domains of known function and may, hence, require a focused analysis.

Inorganic ions are the second largest fraction of known pockets that are not correctly identified via P2Rank (Fig 3). P2Rank may often fail to correctly identify such sites, as they tend to involve a smaller number of residues during binding. P2Rank prediction might be influenced by missing ions or cofactors as it depends on physicochemical properties.

Employing AlphaFill to implant the most likely ions or cofactors into the predicted structures could potentially replace binding site prediction for this compound class [51].

Another type of ligand for which P2Rank struggles to identify the binding sites are carbohydrates, likely due to their complexity in shape, flexibility, and size. This problem has been recently addressed by the development of several specialized prediction tools, such as DeepGlycanSite [52] and PeSTo-Carbs [53]. Specialized binding site prediction tools, such as PocketMiner [54], have also been developed for the detection of cryptic pockets. Cryptic pockets are pockets that are only apparent following a conformational change or upon ligand binding. Complementing the current data set of predicted pockets with binding sites predicted by specialized tools, is a promising route towards investigating the full space of protein binding sites.

Our study once again reveals that binding site comparison remains a challenging task, despite being a subject of research and tool development for many years. While in proteins there are secondary structural elements that allow for a relatively fast and relatively well-defined comparison of proteins, the comparison of protein binding sites is more complex, as they generally do not have a continuous sequence. For a meaningful spatial comparison, rotational and translational invariance is required for binding site comparison tools, such as implemented by DeeplyTough [19]. DeeplyTough compares binding sites in an alignment-free manner by reducing the representation of each pocket to a feature vector. While this captures the overall shape, domain information and physicochemical features of a pocket and allows for fast comparison (simple vector comparison metrics), it is limited in the detection of similar sub-shapes of two binding pockets. Although DeeplyTough was trained to separate dissimilar binding sites by a certain margin with a contrastive loss, the embedding space of DeeplyTough is densely and uniformly populated, making it difficult to obtain a clustering for inter-species comparison as true positives (= similar pockets) as well as false positives (= dissimilar binding sites) are close in the vector space. One reason for that might be pocket types encountered in our study, but that were not found in the limited training data set used by the DeeplyTough developers.

In contrast to alignment-free methods, alignment-based methods, such as ProBiS [30], allow insights into which amino acid residues or surface atoms of the compared pockets are similar and contribute positively to the comparison score. This allows for the easy interpretation of why two binding sites are considered similar or different and, importantly, allows for a sensitive identification of local sub-structural matches. However, these approaches are computationally too expensive at the moment to allow large-scale, inter-species studies on tens of thousands of binding sites as attempted here [29]. Hence, the development of a fast and reliable alignment-based or alignment-free comparison method is still needed to improve inter-species binding site comparison and clustering at scale.

On the other hand, the search space of potentially similar binding sites could be reduced by additional filtering to make alignment-based methods feasible for an all-against-all inter-species comparison. Binding sites that are known to be dissimilar could be pre-filtered by their amino acid composition or their overall physicochemical properties. Binding sites that are known to be similar could be pre-clustered based on the domain to which they belong. This would reduce the number of pairwise comparisons needed, especially for redundant domains, such as the protein kinase domain.

Many of the recently published binding site comparison tools have been evaluated on the ProSPECCTs datasets [28]. ProSPECCTs is an ensemble of seven different data sets of pairs of binding sites with known similarities. In contrast to the aim of this study, the ProSPECCTs datasets do not contain all-against-all comparisons of the respective binding sites. Therefore, the actual performance of the binding sites comparison tools might be overestimated. In addition, the datasets were derived from known protein-ligand complexes. Reim et al. [29] investigated the performance of their tool, SiteMine, for predicted pockets and observed that it is performing worse for some datasets. As predicted pockets consist, on average, of more residues compared to well-annotated sites (Fig 2), they might lose some specificity, and overall physicochemical properties may represent them unspecifically or even incorrectly. This might lead to pockets that are more similar than they actually are. Hence, complicating the binding sites comparison at a large scale. DeeplyTough might lose additional specificity because it is using all surrounding residues of the center of mass of the pocket for

PLOS Computational Biology

embedding, and the usage of predicted pockets instead of known pockets, as they contain, on average, more residues than known pockets.

In addition, using predicted, monomeric protein structures from AlphaFold database results in different challenges that need to be addressed in future research. These challenges include protein-protein interactions, post-translational modifications, and conformational changes that occur upon binding. Currently, the AlphaFold database only includes monomeric structures. However, studies for, e.g., HUMAN and YEAST, suggest that most proteins are at least part of one protein complex [55,56]. Structural predictions might differ between monomer and multimer predictions. Therefore, it would be desirable to investigate potential binding sites from a multimer perspective. Although multimer prediction is challenging due to often unknown stoichiometries and the exploding combinatorics with higher number of chains, reliable prediction of structures for dimers has been achieved by new models, such as AlphaFold-Multimer [57]. The challenge to identify potential interaction partner(s) for each protein of interest through, e.g., protein-protein interaction prediction, remains.

The AF database contains one predicted structure per protein. However, some proteins change their conformation upon binding or have an ensemble of different conformations. These conformational changes are currently not taken into consideration. In addition, studies showed that AF tends to predict the "mean" structure in cases where the apo- and holo-structure of a protein differ [58]. This might influence the prediction and classification of so-called cryptic pockets that are only found in holo-structures. This problem can be tackled by established methods, such as normal mode analysis (NMA), as well as DL-based methods that have been developed recently to overcome this issue [59,60].

Importantly, the AF database does not consider any post-translational modifications, which might influence the structural conformation of proteins. Prominently, protein phosphorylation has been discussed as a modulator of compound binding [61]. A recent, PDB-wide study [62] could show that for phosphorylation, these conformational changes are rather small, with only ~30% of phosphorylation events resulting in an RMSD ≥ 2 Å as compared to the non-phosphorylated structure. However, this number might be too conservative as the PDB database is biased towards less flexible conformations as conformational flexible structures are harder to crystallize. In the same study, the authors could identify a small subset of structures for which phosphorylation is mechanically coupled to functional sites. Extending their work to AF predicted structures, might be a promising avenue of future research.

Another caveat concerns the notion of binding specificity. We approached our study by positing that novel ligands require novel binding sites. However, it is known that binding sites as well as ligands can be promiscuous [16], therefore a one-to-one relationship will not always be true. As a switch from selective to promiscuous binding may involve only small structural changes, functional diversity may grow even in the absence of easily noticeable structural changes.

Lastly, it is known that a substantial portion of proteins and regions within proteins are structurally disordered under physiological conditions [63]. However, it is also known or surmised that these proteins or protein regions do engage in (specific) molecular interactions, including binding to small molecules [64,65]. While structure-based binding site detection in disordered regions are challenging, attempts have been made to apply molecular dynamics simulation to interactions of compounds with disordered protein regions [66], or to predict sequence regions that are likely to interact with other molecules, for example, proteins [67]. Future computational predictions of small molecules with disordered regions of proteins will critically depend on the availability of large-enough experimental datasets as generated by specific NMR protocols [68] and others.

Although some of the challenges mentioned above, such as post-translational modification, are addressed by Alpha-Fold3 [69], structures produced by AF3 are not yet available on such a large scale as they are in the AF database. Hence, they cannot yet be used for a proteome-wide comparison of binding pockets.

## Conclusions

To our knowledge, we present the first large-scale intra- and inter-species study of compound binding sites on protein surfaces using AlphaFold structures to create a map of the pocketome universe. The binding site comparisons conducted for different species using ProBiS reveal that species with larger and more diverse proteomes are not commensurate with

proportionally larger and more diverse pocketomes; instead, a sub-linear relationship between the number of unique pockets vs. unique structures was observed. Our study also documents the vast potential of the new quality and quantity of structural data availability, as well as the power and shortcomings of existing pocket detection and comparison methods.

## Supporting information

**S1 Fig. Violin plots for all species used in the study for the pLDDT of proteins with and without binding sites.** The highest median pLDDT for proteins with and without binding sites was observed for ECOLI, with 93.24 and 90.07, respectively. The lowest median pLDDT of proteins with binding sites was observed for MAIZE with 79.41 and for ORYSJ of proteins without binding sites with 67.05. The median pLDDT values of proteins for the other species range from 80.95 to 85.40 for proteins with binding site and from 68.48 to 75.75 for proteins without any binding site predicted by P2Rank. (TIFF)

**S2 Fig. tSNE plots (perplexity = 10) for all predicted pockets of all 11 species colored by different properties.** (TIFF)

**S3 Fig. tSNE plots (perplexity = 20) for all predicted pockets of all 11 species colored by different properties.** (TIFF)

**S4 Fig. tSNE plots (perplexity = 30) for all predicted pockets of all 11 species colored by different properties.** (TIFF)

**S5 Fig. tSNE plots (perplexity = 40) for all predicted pockets of all 11 species colored by different properties.** (TIFF)

**S6 Fig. tSNE plots (perplexity = 100) for all predicted pockets of all 11 species colored by different properties.** (TIFF)

**S7 Fig.** UMAP plots (n_neighbors = 10) for all predicted pockets of all 11 species colored by different properties. (TIFF)

**S8 Fig.** UMAP plots (n_neighbors = 20) for all predicted pockets of all 11 species colored by different properties. (TIFF)

**S9 Fig.** UMAP plots (n_neighbors = 50) for all predicted pockets of all 11 species colored by different properties. (TIFF)

**S10 Fig.** UMAP plots (n_neighbors = 100) for all predicted pockets of all 11 species colored by different properties. (TIFF)

**S11 Fig.** UMAP plots (n_neighbors = 200) for all predicted pockets of all 11 species colored by different properties. (TIFF)

**S12 Fig. PCA plots for all predicted pockets of all 11 species colored by different properties.** (TIFF)

**S13 Fig. ICA plots for all predicted pockets of all 11 species colored by different properties.** (TIFF)

**S14 Fig. tSNE plot with the first low probability region highlighted in orange.** Pockets that are found in the first low probability region are highlighted in orange. All other pockets are colored in orange (TIFF)

**S15 Fig. tSNE plot colored by species and binned entropy.** (A) tSNE projection colored by different species. All pockets in their respective tSNE embedding space, (B) Binned entropy, S (see Materials and Methods), heatmap plot, capturing for every square bin the species diversity with low entropy, S, values indicating low diversity and high values corresponding to increased diversity. All grid cells containing fewer than 20 pockets are colored black.
(TIFF)

**S1 Table. Additional information for all predicted pockets, including the number of FoldSeek (FS) clusters per species, the ratio of pockets per protein, the number of pockets before filtering, and the number of pockets that did not pass the respective quality check.**
(DOCX)

**S2 Table. Number of unique pockets, singletons, and real clusters with threshold 5 applied to the ProBiS alignment score.**
(DOCX)

## Acknowledgments

We would like to thank Zoran Nikoloski, Ariane Nunes Alves, and Alisdair Fernie for helpful discussions.

## Authors' contributions

Author HZ contributed Conceptualization, Data curation, Formal analysis, Investigation, Methodology, Software development, Computations, Validation, Visualization, Writing – original draft, Writing – review & editing.

Author DW contributed Conceptualization, Formal analysis, Investigation, Methodology, Computation (final version of the Leiden clustering and random controls), Supervision, Validation, Visualization, Writing – original draft, Writing – review & editing

## Author contributions

**Conceptualization:** Hanne Zillmer, Dirk Walther.

**Data curation:** Hanne Zillmer.

**Formal analysis:** Hanne Zillmer, Dirk Walther.

**Investigation:** Hanne Zillmer, Dirk Walther.

**Methodology:** Hanne Zillmer, Dirk Walther.

**Software:** Hanne Zillmer.

**Supervision:** Dirk Walther.

**Validation:** Hanne Zillmer, Dirk Walther.

**Visualization:** Hanne Zillmer, Dirk Walther.

**Writing – original draft:** Hanne Zillmer, Dirk Walther.

**Writing – review & editing:** Hanne Zillmer, Dirk Walther.

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
