## [Decision Letter · Decision Letter 0]

Towards a comprehensive view of the pocketome universe – biological implications and algorithmic challenges.

PLOS Computational Biology

Dear Dr. Walther,

Thank you for submitting your manuscript to PLOS Computational Biology. After careful consideration, we feel that it has merit but does not fully meet PLOS Computational Biology's publication criteria as it currently stands. Therefore, we invite you to submit a revised version of the manuscript that addresses the points raised during the review process.

Please submit your revised manuscript within 60 days Jun 16 2025 11:59PM. If you will need more time than this to complete your revisions, please reply to this message or contact the journal office at ploscompbiol@plos.org. Please include the following items when submitting your revised manuscript:

We look forward to receiving your revised manuscript.

Kind regards,

Turkan Haliloglu

Academic Editor

PLOS Computational Biology

Arne Elofsson

Section Editor

PLOS Computational Biology

**Journal Requirements:**

At this stage, the following Authors/Authors require contributions: Hanne Zillmer, and Dirk Walther. Please ensure that the full contributions of each author are acknowledged in the "Add/Edit/Remove Authors" section of our submission form.

5) Please upload a copy of Figure 12 which you refer to in your text on page 25 line 444. Or, if the figure is no longer to be included as part of the submission please remove all reference to it within the text.

6) We have noticed that you have uploaded Supporting Information files, but you have not included a list of legends. Please add a full list of legends for your Supporting Information files after the references list.

7) Please provide a completed 'Competing Interests' statement, including any COIs declared by your co-authors. If you have no competing interests to declare, please state "The authors have declared that no competing interests exist". Otherwise please declare all competing interests beginning with the statement "I have read the journal's policy and the authors of this manuscript have the following competing interests:"

**Reviewers' comments:**

Reviewer's Responses to Questions

Reviewer #1: This is an important and interesting monumental study of protein binding sites that relies on the library of AlphaFold structures to identify binding sites for a large number of diverse ligands and goes further to compare the characteristics of the sets of binding sites across the broad classes of organisms. The result is an impressive catalog of binding sites of various types across a broad range of organisms, which presents results important for evolutionary information about how binding sites have evolved. Overall, the study is extremely impressive in its breadth and reliance on many different methods. The discussion at the end includes some useful general discussions about factors that could be problematic but lacks some more specific details. Some quantitative testing of the sensitivity of the cutoffs that have been used would be helpful to the reader. The cutoff value of 0.5 used on P2Rank values needs some justification or evaluation of how results change with different cutoff values. Likewise, there is some further information needed regarding the cutoff red line in Figure 1.

The predicted sets of protein structures from AlphaFold have proven to be extraordinarily informative and are now being used for many different purposes. There are competitive predicted protein structures from other researchers such as RosettaFold and ESMFold but the limited number of comparisons have generally shown that the AlphaFold structures are better. There are also sets of experimentally determined structures that could have been included, but these are woefully incomplete.

One major problem with the AlphaFold protein structures is that many of the structures have long external unfolded segments or segments with only secondary structures, which many researchers are interpreting to be disordered parts or membrane anchors. In principle, these could be affecting some binding sites, but at present I don’t think there is anything reliable or any systematic way to treat these pieces. We and others are simply cutting them off.

In Figure 2, the label derivates should be derivatives.

In the discussion regarding domains, I believe the word domain-no-domain should be domain-to-domain. Also, since the most common motions in proteins are between domains, these cases may be the most affected by the dynamics of the structures.

Reviewer #2: The presented manuscript addresses an intriguing question: whether binding pockets differ across species and how genome size correlates with the number of predicted pockets. To investigate this, the authors analyze AlphaFold-predicted proteomes from several species, apply the P2Rank pocket prediction tool to identify potential binding sites, and use two different similarity assessment methods to evaluate pocket variability both within and between species.

Overall, the manuscript is relatively easy to follow, though some sentences are overly complex, which occasionally hinders comprehension (see comments below). I find the idea of exploring interspecies pocket variation highly interesting, and the scope of the work is solid. However, I have several methodological concerns, outlined below.

Major comments:

- I do not fully understand the rationale for removing proteins with fewer than 100 amino acids. According to [1], about 20% of human proteins are shorter than 100 amino acids (see Fig. 3c). Even if proteins with ion-only binding sites are excluded, this still leaves a substantial number of proteins, likely around 5% (though it is difficult to read from the graph). This exclusion seems arbitrary and may result in a loss of valuable data.

- Pg 6: "If two pockets overlapped, only the pocket that was predicted with a higher probability was kept." -It seems that the situation is not quite right, especially considering that P2Rank predicted pockets, as stated by the authors, are larger than the actual pockets. This implies that two pockets overlapping by a single residue might actually be two distinct pockets. Focusing on only one of them could result in the loss of many valid predictions (though it's unclear how often this happens). Additionally, if we examine Figure 5D from reference [1], we can see that there are indeed several binding sites in the human proteome where residues overlap. In general, the pockets can be quite close to each other, as demonstrated in Figure 5C..

It was difficult for me to decode the sentence capturing the main conclusion: "Thus, larger proteomes harbor less than proportionally more different binding sites than species with smaller proteomes." I recommend rephrasing this.

- The manuscript repeatedly mentions that P2Rank's predicted pockets are much larger than the observed pockets, which improves performance, as reported in [1]. Why wasn’t this mode of prediction used in this study?

- Pg 14: P2Rank, like many other pocket prediction tools, was not trained on ion-binding data, which limits its performance in this area. This point deserves more discussion.

- Tab 4 - I believe both t-SNE and UMAP will produce different embeddings when embedding the data in one dimension versus two dimensions. Therefore, it doesn't make sense to use tSNE1 and tSNE2 independently. If the goal is to correlate t-SNE with a specific property, the data should be projected into one dimension, which would yield different results than projections into two dimensions and the project onto individual axes.

- Replicability: I believe the results are not replicable because the authors only provide the UMAP embedding and fail to include the data, predictions, or code for analysis.

Minor comments:

- MMseqs2 was used with default settings. What is the default similarity threshold for the version of MMseqs2 used in this study? This should be explicitly stated to provide clarity on the data.

-Pg 6: "pockets that include between 4 and 31 residues were considered further" - This seems rather arbitrary. Why was this specific threshold chosen?

- Pg8, ln173 - Again, the choice of perplexity for t-SNE seems arbitrary. It is well known that the choice of perplexity significantly impacts clustering. How sensitive are the results to variations in this parameter?

- The authors mention that ProBis was not used for inter-species comparison due to computational resource limitations. Would it be feasible to run the analysis on high-performance computing (HPC) systems? If so, the resource requirements should be quantified. The manuscript does not provide enough information on how resource-intensive ProBis is.

- Related to previous - The authors suggest that DeeplyTough is inferior to ProBis. This claim should be supported by a relevant reference.

- Pg 8 - I was unfamiliar with this term, so I looked it up. It appears that steerable CNNs are equivariant neural networks (i.e., invariant to rotation). However, I do not understand why data augmentation is needed to achieve rotational invariance, as mentioned in line 161.

- Pg 9 "The PAE 194 values for each protein of interest were obtained from the AF database, and the pLDDT values were extracted from the corresponding protein structures" - pLDDT also come from the AF database. Sure, they are stored in the PDB files, but they are in the AF database nevertheless.

- Pg 9: "were calculated for all obtained physicochemical and AF-quality pocket descriptors correlated with each dimension of the embedding vector, performed for all dimensionality reduction methods." - That reads strange.

- Pg 9: "As DeeplyTough’s embedding space for all binding sites was too densely populated to obtain a meaningful clustering" - I don't think "dense" is an issue for clustering, uniform distribution of the data (which used to be the case when increasing the dimension of data) would be...

-Pg 10: "which has about the same number of predicted 236 pockets as ARATH although it has about 6,000 (~24%) more predicted protein structures than 237 ARATH" - Is this after filtering out small structures?

-Fig1 - Why showing PAE and not pLDDT?

- Fig.1. - Why are the PAE values squared considering they are positive?

- Ln255: "there are no significant differences of the estimated prediction quality" - statistically? If not, it should be possible to do statistical significance testing

- Tab. 2: I believe that majority of the caption should be part of the text.

- Pg. 17: "This implies that singletons of low probability potentially are pockets that are not actually involved in any ligand binding." - As noted on the PrankWeb website, the probabilities of PrankWeb are calculated on existing structures, i.e., how well the raw score correlates with observed binding sites. That could lead to seeing low scores in non-homologous pockets.

- Pg 18, Ln353/354: N_pocket/N_FS - is this aver or mean. If it is average, could it be driven by outliers?

- Pg 26: It would be fair to say that clustering is driven by the features of the objects. Different embedding methods will likely result in different clustering, as they view the original data differently. Therefore, the clustering plot represents the feature extraction method’s perspective on the data rather than the underlying data itself, unless there is strong evidence that the feature extraction method truly captures the underlying data. This should be discussed in more detail.

Typos:

Ln237 - number pockets -> number of pockets

Ln337 - processes -> processed

Ln346 - proteomes -> proteins (?)

Ln367 - low -> law

[1] Utgés, Javier S., and Geoffrey J. Barton. "Comparative evaluation of methods for the prediction of protein–ligand binding sites." Journal of Cheminformatics 16.1 (2024): 126.

Reviewer #3: This manuscript presents a large-scale analysis of predicted small-molecule binding pockets (the "pocketome") in protein structures from eleven species, using AlphaFold-predicted 3D models. The authors applied a computational pocket detection method across the full proteomes of selected species from different kingdoms of life, with the aim of characterizing pocket diversity and identifying evolutionary patterns. A notable observation is the identification of a sub-linear scaling law relating the number of unique binding pockets to the number of unique protein structures within a species, suggesting that organisms with more diverse proteomes may not necessarily have proportionally more diverse pocketomes.

While the study is technically competent and the topic remains broadly relevant to computational structural biology, the principal concern is the limited novelty and incremental contribution of the findings. The observed scaling law is interesting, but the concept of saturation or sub-linearity in structure-function space is not new and has been described in earlier literature. Additionally, the idea that larger proteomes may include more redundancy (or fewer novel binding sites per structure) is intuitively expected and has been touched on in previous proteome- and ligand-binding studies.

Moreover, large-scale surveys of binding pockets across species, including those based on experimental structures, have been published more than a decade ago—for instance, the study titled “A Comprehensive Survey of Small-Molecule Binding Pockets in Proteins” (2012) and others that followed have examined very similar questions, often with greater scope or methodological innovation. While the current manuscript benefits from the structural coverage provided by AlphaFold, the core scientific insights do not significantly advance the state of knowledge or introduce a clearly novel conceptual framework.

The conclusions drawn in this study are relatively narrow in scope: the scaling relationship and speculation on evolutionary implications are intriguing but not fully developed or deeply explored. Additionally, the dependence on predicted rather than experimentally validated structures may also limit the interpretability of the results, especially when drawing functional or evolutionary conclusions about binding sites.

PLOS Computational Biology aims to publish work of broad interest and exceptional significance within the computational biology field. While this study may be of value to researchers working specifically on protein structure prediction or ligand binding site analysis, the overall contribution does not rise to a level of impact or novelty that aligns with the journal mission. The manuscript may be more suitable for a specialized journal focused on structural bioinformatics or protein annotation.

In summary, the manuscript is well-structured and presents a technically sound analysis, but the lack of conceptual innovation and limited advancement beyond prior work reduce its significance for a general computational biology audience.

**Have the authors made all data and (if applicable) computational code underlying the findings in their manuscript fully available?**

Reviewer #1: Yes

Reviewer #2: **No: ** The authors only provide the UMAP embedding and fail to include the data, predictions, or code for analysis.

Reviewer #3: None

PLOS authors have the option to publish the peer review history of their article (what does this mean? ). If published, this will include your full peer review and any attached files.

**Do you want your identity to be public for this peer review?** For information about this choice, including consent withdrawal, please see our Privacy Policy .

Reviewer #1: No

Reviewer #2: No

Reviewer #3: No

**Figure resubmission:**

**Reproducibility:**



---

## [Decision Letter · Decision Letter 1]

Dear Professor Walther,

We are pleased to inform you that your manuscript 'Towards a comprehensive view of the pocketome universe – biological implications and algorithmic challenges.' has been provisionally accepted for publication in PLOS Computational Biology.

Also, if possible, please consider the minor comments by one of the reviewers.

Best regards,

Turkan Haliloglu

Academic Editor

PLOS Computational Biology

Arne Elofsson

Section Editor

PLOS Computational Biology

Reviewer's Responses to Questions

**Comments to the Authors:**

Reviewer #1: This paper seeks to examine the pocketome (space of equivalent binding pockets for several classes of ligands) and to find an association across species between complexity of proteome and number of unique pockets. The authors note that Alphafold now provides much more data than could have been analyzed previously, allowing for this large-scale study.

The revisions are significantly more rigorous and more complete. Overall, this is a quite thorough article that has an interesting finding on the scaling of binding site diversity with respect to proteome.

Two references that might be added are:

[1] Y. Liu, P. Li, S. Tu and L. Xu, "RefinePocket: An Attention-Enhanced and Mask-Guided Deep Learning Approach for Protein Binding Site Prediction," IEEE/ACM Transactions on Computational Biology and Bioinformatics, vol. 20, no. 5, pp. 3314-3321, 2023.

[2] O. Vural and L. Jololian, "Machine learning approaches for predicting protein-ligand binding sites from sequence data," Frontiers in Bionformatics, vol. 5, 2025.

One minor comment - why were Archaea proteins not considered?

Reviewer #2: All my comments have been sufficiently addressed.

**Have the authors made all data and (if applicable) computational code underlying the findings in their manuscript fully available?**

Reviewer #1: Yes

Reviewer #2: Yes

PLOS authors have the option to publish the peer review history of their article (what does this mean? ). If published, this will include your full peer review and any attached files.

**Do you want your identity to be public for this peer review?** For information about this choice, including consent withdrawal, please see our Privacy Policy .

Reviewer #1: **Yes: ** Robert Jernigan

Reviewer #2: No

---

## [Editor Report · Acceptance letter]

PCOMPBIOL-D-25-00258R1

Towards a comprehensive view of the pocketome universe – biological implications and algorithmic challenges.

Dear Dr Walther,

I am pleased to inform you that your manuscript has been formally accepted for publication in PLOS Computational Biology. Your manuscript is now with our production department and you will be notified of the publication date in due course.

With kind regards,

Zsuzsanna Gémesi
